# Autoregressive Search Engines:
# Generating Substrings as Document Identifiers

**Michele Bevilacqua**[1,2]  **Giuseppe Ottaviano**[2]  **Patrick Lewis**[2]
**Wen-tau Yih**[2]  **Sebastian Riedel**[2,3]  **Fabio Petroni**[2]
[1]Sapienza University of Rome  [2]Meta AI  [3]University College London

## Abstract

Knowledge-intensive language tasks require NLP systems to both provide the correct answer and retrieve supporting evidence for it in a given corpus. Autoregressive language models are emerging as the de-facto standard for generating answers, with newer and more powerful systems emerging at an astonishing pace. In this paper we argue that all this (and future) progress can be directly applied to the retrieval problem with minimal intervention to the models' architecture. Previous work has explored ways to partition the search space into hierarchical structures and retrieve documents by autoregressively generating their unique identifier. In this work we propose an alternative that doesn't force any structure in the search space: using all ngrams in a passage as its possible identifiers. This setup allows us to use an autoregressive model to generate and score distinctive ngrams, that are then mapped to full passages through an efficient data structure. Empirically, we show this not only outperforms prior autoregressive approaches but also leads to an average improvement of at least 10 points over more established retrieval solutions for passage-level retrieval on the KILT benchmark, establishing new state-of-the-art downstream performance on some datasets, while using a considerably lighter memory footprint than competing systems. Code and pre-trained models are available at `https://github.com/facebookresearch/SEAL`.

## 1 Introduction

Surfacing knowledge from large corpora is a crucial step when dealing with knowledge intensive language tasks [Levy et al., 2017, Dinan et al., 2019, Elsahar et al., 2018, Petroni et al., 2021], such as open-domain question answering [Voorhees et al., 1999, Joshi et al., 2017, Yang et al., 2018, Kwiatkowski et al., 2019] and fact checking [Thorne et al., 2018]. A popular paradigm to approach such tasks is to combine a search engine with a machine reader component. The former retrieves relevant context, usually in the form of short passages, which the latter then examines to produce answers [Chen et al., 2017, Lewis et al., 2020, Izacard and Grave, 2021].

In recent years we have witnessed a surge of research and development in autoregressive language models [Radford et al., 2019, Lewis et al., 2019, Raffel et al., 2019, Brown et al., 2020, Rae et al., 2021, Artetxe et al., 2021, Smith et al., 2022], with ever increasing size and natural language understanding (NLU) capabilities. Such models are currently the de-facto implementation of the machine reader component in retrieval-reader architectures, and have contributed to rapid progress on a wide range of benchmarks [Joshi et al., 2017, Kwiatkowski et al., 2019, Petroni et al., 2021]. However, these tremendous advances in aggressive modelling has yet to bring similar transformational changes in how retrieval is approached.

Transferring the NLU capabilities of modern autoregressive models to retrieval is non-trivial. Some works have demonstrated that knowledge stored in the parameters of these models can be retrieved to some extend by directly generating evidence given a query [Petroni et al., 2019, 2020, Roberts

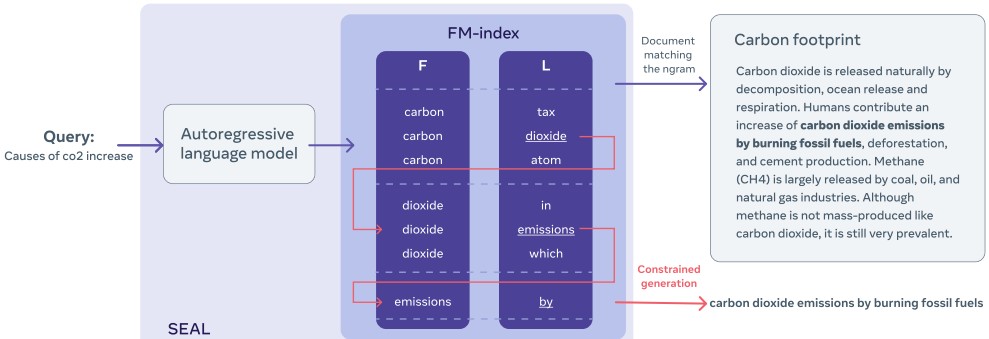

Figure 1: High-level SEAL architecture, composed of an autoregressive LM paired with an FM-Index, for which we show the first (F) and last (L) columns of the underlying matrix (more details in Sec 3.1). The FM-index constrains the autoregressive generation (*e.g.*, after *carbon* the model is contrained to generate either *tax*, *dioxide* or *atom* in the example) and provides the documents matching (*i.e.*, containing) the generated ngram (at each decoding step).

et al., 2020]. However, such approaches have been shown to be unreliable because of their tendency to hallucinate non-factual content [Massarelli et al., 2019, Metzler et al., 2021, Ji et al., 2022]. To alleviate this issue, previous work proposes to only use generation for query expansion in traditional search engines [Mao et al., 2021], but these solutions don't exploit the full potential of autoregressive architecture, such as word order sensitivity and conditional probability modeling, and still lag behind vector-based approaches [Karpukhin et al., 2020].

Recently, another line of work has investigated using autoregressive language models to generate identifier strings for documents as an intermediate target for retrieval, such as Wikipedia page titles [De Cao et al., 2021b], or root-to-leaf paths in a hierarchical cluster tree [Tay et al., 2022]. Employing identifiers, rather than generating evidence directly, induces structure in the search space, (*i.e.*, index documents by their title or their cluster tree) which can be easier to memorize, learn, and retrieve from, than full unstructured passages. Moreover, it is relatively easy to constrain decoding with a prefix tree "index" so that only valid identifiers are generated. As a downside, if appropriate metadata (*e.g.*, titles) are not available, one needs to create the identifiers, hence the structure (*e.g.*, with hierarchical clustering), which has not been thoroughly evaluated on a large-scale benchmark.

In this work, we propose a solution that does not force any structure in the search space, but rather uses all the ngrams occurring in a document as its identifiers. Concretely, we introduce **S**earch **E**ngines with **A**utoregressive **L**Ms (SEAL), a retrieval solution that combines an autoregressive model, *i.e.*, BART [Lewis et al., 2019], with a compressed full-text substring index, *i.e.*, the FM-Index [Ferragina and Manzini, 2000] — see Figure 1 for an high-level overview. This configuration comes with a twofold benefit: i) we can constrain BART's generations with the FM-Index, hence preventing the generation of invalid identifiers (*i.e.*, ngrams not occurring in any document); ii) the FM-Index provides information on all documents in the corpus containing a specific ngram (for every decoding step), thus allowing to retrieve them. This setup allows SEAL to generate *any span* from *any position* in the corpus, without needing to explicitly encode all substrings in a document. Moreover, we design a novel scoring function to intersect the results of multiple ngrams combining LM probabilities with FM-index frequencies (*i.e.*, number of occurrences of the ngram in the whole corpus).

Our experimental evaluation shows that SEAL matches or outperforms recent retrieval solutions (including autoregressive ones) on Natural Questions [Kwiatkowski et al., 2019], while requiring substantially less memory ($\sim$2 to 7 times smaller in footprint). Moreover, SEAL's intersection formulation improves the state-of-the-art on passage-level retrieval by more than 10 points on the KILT benchmark [Petroni et al., 2021], contributing in establishing new state-of-the-art downstream results on multiple datasets when paired with existing reader technologies.

## 2 Related Work

One way to approach retrieval with autoregressive models makes use of unique identifiers, *i.e.*, string pointers to documents that are in some way easier to generate than the full document itself [De Cao

et al., 2021b, Tay et al., 2022]. In our work the identifiers are generated ngrams, which do not necessarily occur in just one document. Note that the idea of using phrases to retrieve passages has shown promise already in the context of dense retrieval [Lee et al., 2021a,b].

Our approach is quite conceptually distinct from document [Nogueira et al., 2019, Nogueira and Lin, 2021] and query expansion approaches [Mao et al., 2021], that use autoregressive models to boost search, but still rely on black-box systems like BM25 [Robertson and Zaragoza, 2009] for the retrieval: in our work the boundary between generation and retrieval is blurred, since we generate *grounded* passage spans. The same reasoning applies to models that generate queries for web search engines [Komeili et al., 2021, Shuster et al., 2022, Nakano et al., 2021, Lazaridou et al., 2022].

Virtually all modern approaches to string-matching-based retrieval use bag-of-words representations. Many recent works propose learned contextualized weighting for both queries and documents terms [Dai and Callan, 2019, Gao et al., 2021, Lin and Ma, 2021, Mallia et al., 2021, Dai and Callan, 2020, Bai et al., 2020, Zhao et al., 2021, Formal et al., 2021b,a]. Many of these methods can also weigh terms that are not present in the query, addressing so-called vocabulary mismatch. In contrast, SEAL generates (and scores) ngrams of arbitrary size. These approaches are partly orthogonal to SEAL, as many of the proposed techniques could be used to rescore higher-order ngrams.

Finally, a connected strand of research is that of query likelihood models, which use autoregressive models to (re)rank passages according to the probability $P(q|p)$ of a query $q$ given the passage $p$ [Nogueira dos Santos et al., 2020, Zhuang and Zuccon, 2021, Lesota et al., 2021, Sachan et al., 2022]. In our case, the autoregressive architecture models the likelihood of an ngram given the query, *i.e.*, $P(n|q)$.

## 3 Background

In retrieval, the automatic system is required to return an ordered list of documents $d_1, d_2, \ldots, d_n$ from a retrieval corpus $\mathcal{R}$, given a query $q$. Both queries and documents are texts, *i.e.*, lists of tokens $\langle t_1, t_2, \ldots, t_N \rangle$, where each token $t$ is drawn from a vocabulary $V$. A span of tokens in a text is called an ngram; ngrams of size 1 are known as unigrams. We denote with $F(n, \mathcal{R})$ the frequency of an ngram $n$ in $\mathcal{R}$, *i.e.*, the total number of times it appears in the whole retrieval corpus.

### 3.1 The FM-Index

Our method requires a data structure that can support the efficient identification of occurring substrings to guarantee that all decoded sequences are located somewhere in the retrieval corpus. Moreover, to perform retrieval, we require the ability to identify which documents the generated ngrams appear in. Neither inverted indices (which have no efficient way to search for phrases of arbitrary length), nor prefix trees (which would force us to explicitly encode all $k$ suffixes in a document), are viable options. The core data structure that satisfies our requirements is the FM-index [Ferragina and Manzini, 2000], *i.e.*, a compressed suffix array that, as a self-index, requires no additional storage for the original text. FM-index space requirements are linear in the size of the corpus, and, with small vocabularies such as those used by modern subword-based language models, is thus usually *significantly smaller* than the uncompressed corpus. The FM-index can be used to count the frequency of any sequence of tokens $n$ in $O(|n|\log|V|)$, *i.e.*, independently from the size of the corpus itself. For constrained decoding, the list of possible token successors can be obtained in $O(|V|\log|V|)$. Internally, the FM-index relies on the Burrows-Wheeler Transform [Burrows and Wheeler, 1994], or *BWT*, an invertible transformation that permutes a string to make it easier to compress, defined as follows: all the rotations of the string are sorted lexicographically and laid out in a matrix; the last column of the matrix is the strings's BWT.[1] For example, given the string $CABAC$, the corresponding matrix would be:

$$
\begin{array}{cccccc}
\mathbf{F} & & & & & \mathbf{L} \\
\$^6 & C & A & B & A & C^5 \\
A^2 & B & A & C & \$ & C^1 \\
A^4 & C & \$ & C & A & B^3 \\
B^3 & A & C & \$ & C & A^2 \\
C^5 & \$ & C & A & B & A^4 \\
C^1 & A & B & A & C & \$^6
\end{array}
$$

---

[1]Since our corpus contains multiple documents, we concatenate them with a separator token.

where $ is a special end-of-string token. The first (**F**) and last (**L**) columns are the only ones that will be explicitly stored in the FM-index; **F** is just an array of runs (*i.e.*, sequences of repeated tokens), due to the rotations being sorted, so it can be represented with one count for each alphabet symbol; **L**, the string's BWT, will be stored in a data structure known as the Wavelet Tree [Grossi et al., 2003], which allows efficient rank-select-access queries, while exploiting the compressibility induced by the transformation. FM-indices have the useful property that for each symbol, the relative rank stays the same: that is, the $i$th occurrence of a symbol $\sigma$ in **F** points to the same location in the corpus of the $i$th occurrence of $\sigma$ in **L**. Thanks to this property, we can locate any string $\langle \sigma_1, \sigma_2, \ldots, \sigma_n \rangle$ in the index by starting from $\sigma_n$ and going backwards. First, we select the contiguous range of rows corresponding to the symbol $\sigma_n$ in **F**, then we check the ranks of the first and last occurrences of $\sigma_{n-1}$ in the same range of rows in $L$. We use the ranks to select a new, smaller or equal range of rows looking up the symbol $\sigma_{n-1}$ in $F$. The procedure can be applied iteratively to find ngrams of any size.

## 4 Method

In our retrieval methodology, SEAL, we generate multiple ngrams, conditioning on a query. The ngrams are then used to find the documents they appear in within the corpus, which are then returned to the user. In Figure 1 we show this process at a high-level. We use our indexing structure, *i.e.*, the FM-index to do constrained decoding so that each ngram occurs at least once in the retrieval corpus. Jointly, we use the FM-index to efficiently find matching documents. Documents are ranked using the scores of the generated ngrams.

**Autoregressive Retrieval**   We generate ngrams identifiers with constrained beam search, using the FM-index to identify the set of possible next tokens in at most $O(|V|\log|V|)$: tokens corresponding to unattested continuations are blocked by masking the logit to $-\infty$. As a result, after a single decoding pass, we get a set of ngrams ($K$), along with their autoregressively-computed probabilities according to the model. It is also trivial to find the positions in the corpus where the decoded ngrams appear, as constrained decoding already requires selecting the relevant range of rows in the FM-index. Note that autoregressive scoring entails monotonically decreasing scores—any string will be assigned a lower probability than any of its prefixes. To address this issue, we use fixed-length ngrams. Each document is assigned the score ($P(n|q)$) of its most probable decoded occurring ngram, *i.e.*, the probability of the ngram as autoregressively computed by the decoder, conditioned on the encoder input We refer to this as the **LM** scoring.

**Factoring in FM-index frequencies**   To counterbalance the monotonic probability decrease, we integrate in scoring unconditional ngram probabilities, computed as normalized index frequencies:

$$P(n) = \frac{F(n, \mathcal{R})}{\sum_{d \in \mathcal{R}} |d|} \tag{1}$$

This also enables us to promote *distinctive* ngrams, *i.e.*, those that have high probability according to the model and low probability according to the FM-index. We take inspiration from the theory behind TF-IDF and BM25 [Robertson and Zaragoza, 2009] and use the following scoring function:

$$w(n, q) = \max\left(0, \log \frac{P(n|q)(1 - P(n))}{P(n)(1 - P(n|q))}\right) \tag{2}$$

This formulation addresses the problem of length, as the unconditional probability of an ngram will also be equal or lower than that of any of its prefixes. To make better use of the computational resources, we slightly modify the beam search implementation to keep track of all the partially decoded sequences that have been considered. Thanks to this, we score a larger number of ngrams than the size of the beam. We refer to this formulation as the **LM+FM** scoring.

**An Intersective Scoring for Multiple Ngrams**   One problem with the previous scoring formulations is that it is impossible to break ties among documents whose highest scoring ngram is the same, as they receive exactly the same score. Moreover, it might be difficult to capture all relevant information within a document by considering only a single ngram, for instance when salient ngrams are non-contiguous (*e.g.*, separated by unrelated text). To address these issues we propose a novel scoring

formulation that aggregates the contribution of multiple ngrams contained in the same document. To avoid repeated scoring of overlapping ngrams, for each document $d \in \mathcal{R}$ we only consider a subset of the generated ngrams $K^{(d)} \subset K$. An ngram $n$ belongs to $K^{(d)}$ if there is at least one occurrence of $n$ in $d$ that does *not* overlap in the corpus with an occurrence of another ngram $n'$ such that a) $n' \in K^{(d)}$ b) $w(n', q) > w(n, q)$. The document-level score, then, is the weighted sum of all ngrams in $K^{(d)}$:

$$W(d, q) = \sum_{n \in K^{(d)}} w(n, q)^\alpha \cdot \text{cover}(n, K^{(d)}) \tag{3}$$

where $\alpha$ is a hyperparameter and the coverage weight $\text{cover}(n, K)$ (controlled by the second hyperparameter $\beta$) is a function of how many ngram tokens are not included in the coverage set $C(n, K) \subset V$, *i.e.*, the union of all tokens in ngrams with a higher score. We define this coverage weight as follows:

$$\text{cover}(n, K) = 1 - \beta + \beta \cdot \frac{|\text{set}(n) \setminus C(n, K)|}{|\text{set}(n)|} \tag{4}$$

where $\text{set}(n)$ is the set of all tokens in $n$. The purpose of the coverage weight is to avoid the overscoring of very repetitive documents, where many similar ngrams are matched. Note that by saving the probability distribution at the first decoding step we can compute scores for all unigrams with no additional forward pass. We refer to this last approach, which can be thought of as a higher-order generalization of the bag-of-words assumption, as the **LM+FM intersective** scoring.

## 5 Experimental Setting

Our experimental setting evaluates SEAL on English knowledge-intensive NLP tasks. Each considered dataset is a collection of queries, each of which can be answered by looking for piece(s) of evidence in the corpus. We consider both an in vivo evaluation, in which we assess the model by looking at how well the document ranking matches with the ground truth, and, in addition, we perform a downstream evaluation, in which we feed the retrieved documents to a trained reader, that uses the documents to generate the answer.

### 5.1 Data

**Natural Questions**    Natural Questions (NQ) is dataset containing query-document pairs, where the query is a question (*e.g.*, "who wrote photograph by ringo starr"), and the document is a Wikipedia page, in which a span is marked as an answer [Kwiatkowski et al., 2019]. We experiment on both the customary retrieval setup used by, among others, Karpukhin et al. [2020] and Mao et al. [2021], and the substantially different setup used by Tay et al. [2022]. We refer to these two settings as, respectively, **NQ** and **NQ320**$k$. In NQ, retrieval is performed on an entire Wikipedia dump, chunked in around 21M passages of 100 tokens. Performance is measured as accuracy@$k$, *i.e.*, the fraction of instances for which at least one of the top-$k$ retrieved passages contains the answer. NQ320$k$ is a much more restricted setting, in which the retrieval set is limited to the union of all ground truth document in the training, dev or test set. Different revisions of the same Wikipedia page count as different documents. Note that the exact splits used by Tay et al. [2022], the retrieval corpus and the preprocessing code have not been yet released at the time of writing. Therefore, we have tried to replicate the setting as closely as possible, but the exact numbers are not precisely comparable with those reported in the original paper. In NQ320$k$, performance is measured as hits@$k$, i.e, the fraction of instances for which at least one of the top-$k$ retrieved passages is in the ground truth.

**KILT**    is a comprehensive benchmark collecting different datasets including question answering, fact checking, dialogue, slot filling, and entity linking [Petroni et al., 2021]. All these tasks are solvable by retrieving information from a unified corpus — a Wikipedia dump. In KILT, the evidence is usually the paragraph that contains the answer. Following Maillard et al. [2021], we have re-chunked KILT's retrieval corpus, which is originally paragraph-based, in around 36M passages of 100 tokens. We do not use the entity linking and ELI5 KILT tasks, where a ground truth passage is not provided in the training set. KILT's retrieval performance is measured with R-precision, a precision-oriented measure that considers only gold documents as correct answers, not just any document containing the answer. R-precision can be computed at either passage level or at page level.

Table 1: Language model and index size on Natural Questions (around 21M passages). SEAL's index is ~1.5 times smaller than uncompressed plain text.

| System | Model Params | Size | Index Params | GPU? |
|--------|--------------|------|--------------|------|
| *plain text* | - | 13.4GB | - | - |
| DPR | 220M | 64.6 GB | 16.1B | ✓ |
| BM25 | - | 18.8 GB | - | ✗ |
| GAR | 406M | 18.8 GB | - | ✗ |
| DSI-BART | 406M | - | - | - |
| SEAL | 406M | 8.8GB | - | ✗ |

## 5.2 SEAL configuration

**Training**   We finetune BART large [Lewis et al., 2019] to generate ngrams of length $k = 10$ from the ground truth document. Since there are $|d| - k$ ngrams in a document $d$, we sample (with replacement) 10 ngrams from it, biasing the distribution in favor of ngrams with a high character overlap with the query. We also add the title of the document to the set of training ngrams. To expose the model to more possible pieces of evidence, we also add different "unsupervised" examples for each document in the retrieval corpus to the training set. In each of these examples the model takes as input a uniformly sampled span from the document, and predicts either another sampled span, or the title of the page. We append special tokens to the input to signal to the model a) whether the pair comes from the supervised or unsupervised training pairs (in the same spirit as the co-training task prompts used by Tay et al. [2022]) b) whether a title or span is expected as output. On KILT we train SEAL on all datasets at once. We report training and inference hyperameters in Appendix (§A).

**How to choose ngrams**   In preliminary experiments, we have found that training SEAL with ngrams of fixed length sampled uniformly from the training chunk did not lead to good performances. Instead, we have biased the ngram distribution towards ngrams that were relevant to the query. As a simple proxy for relevance, we have use the Levenshtein (character-based) distance between the query and the ngram overlap. The final distribution to sample from is defined by the following formula:

$$\frac{e^{L(q, d_{i:i+k})/\tau}}{\sum_{j=1}^{|d|-k+1} e^{L(q, d_{j:j+k})/\tau}} \tag{5}$$

where $L$ is the Levenshtein distance and $\tau$ ($= 1.5$ in our experiments) is a temperature parameter, controlling the peakiness of the distribution.

**Index**   We use the C++ FM-index implementation in `sdsl-lite`. While the FM-index construction (which requires a sort of all rotations) takes around 6 hours in our single-threaded implementation, parallel algorithms are available [Labeit et al., 2017]. Each document is encoded as the subword tokenization of the concatenation of the title and the passage, separated by a special token. We report in Table 1 the index statistics for Natural Questions. As can be seen, SEAL's FM-index is more than 7 times lighter compared to DPR's full document embeddings for exact inner product search, and needs neither a GPU for search on top of that, nor separate storage for the text itself. While vector compression methods can reduce dense retrievers' index size, this still comes at the expense of performance [Yamada et al., 2021, Lewis et al., 2021a]. In addition, our the size of our index is less than 50% of that of the well-optimized Lucene BM25 index used by `pyserini`, but also roughly 65% of the uncompressed plain text itself.

## 5.3 Retriever Baselines

We compare SEAL against well-established systems in the literature on each benchmark. On NQ and NQ320$k$ we also compare against our BART-based replication of DSI [Tay et al., 2022, **DSI-BART**].[2] On NQ320k, a page-level benchmark, we include our own replication of GENRE [De Cao et al., 2021b]. Unless otherwise specified, we use `pyserini` to compute the BM25 baseline. For other systems, we either take figures from the literature, or use publicly released model predictions.

---

[2]We report reproduction details in Appendix (§B).

Table 2: Results on NQ320$k$. Reporting hits@1 and hits@10. Best in bold.

| System | hits@$k$ | |
|---|---|---|
| | **1** | **10** |
| BM25 (`gensim`) | 15.3 | 44.5 |
| BM25 | 22.7 | 59.0 |
| DSI-BART | 25.0 | 63.6 |
| GENRE | **26.3** | 71.2 |
| SEAL (LM, $|n| = 3$) | 21.3 | 66.5 |
| SEAL (LM, $|n| = 4$) | 22.2 | 68.2 |
| SEAL (LM, $|n| = 5$) | 22.6 | 68.7 |
| SEAL (LM+FM) | 25.3 | 72.0 |
| SEAL (LM+FM, intersect.) | **26.3** | **74.5** |

Table 3: Retrieval results on the NQ test set. Column blocks (left to right): retrieval results (accuracy@5/20/100); retrieval results on the test splits of Lewis et al. [2021b], partitioned according to whether the query/answer is a paraphrase of one in the training set; downstream performances (exact match). Except for Izacard and Grave [2021], all downstream results are computed with the same FiD reader trained on DPR predictions. Best in bold.

| System | accuracy@$k$ | | | Overlap? (A@100) | | | | EM |
|---|---|---|---|---|---|---|---|---|
| | **5** | **20** | **100** | ans. ✓ | ✗ | ques. ✓ | ✗ | |
| BM25 | 43.6 | 62.9 | 78.1 | 82.9 | 70.1 | 80.9 | 76.6 | 40.4 |
| DPR [Karpukhin et al., 2020] | **68.3** | **80.1** | 86.1 | 91.4 | 76.8 | 93.2 | 83.2 | 47.2 |
| GAR [Mao et al., 2021] | 59.3 | 73.9 | 85.0 | **91.6** | 74.4 | **94.1** | 80.4 | 46.2 |
| DSI-BART | 28.3 | 47.3 | 65.5 | 77.8 | 44.2 | 84.9 | 57.7 | 31.4 |
| Izacard and Grave [2021] | - | - | - | - | - | - | - | **48.2** |
| SEAL (LM, $|n| = 5$) | 40.5 | 60.2 | 73.1 | 82.2 | 57.1 | 85.2 | 64.9 | 36.0 |
| SEAL (LM+FM) | 43.9 | 65.8 | 81.1 | 86.9 | 70.9 | 89.5 | 78.1 | 42.9 |
| SEAL (LM+FM, intersective) | 61.3 | 76.2 | **86.3** | 91.2 | **77.7** | 93.2 | **84.1** | 48.0 |

## 5.4 Reader

For downstream results, we use the Fusion-in-Decoder abstractive reader [Izacard and Grave, 2021], which takes in the query along with 100 contexts and produces a task-specific answer. We train FiD on training set predictions.

## 6 Results

**NQ320$k$** We report results on NQ320$k$ in Table 2. SEAL outperforms BM25 and DSI-BART in hits@10 in all its formulations. When taking into account ngram frequencies (*i.e.*, LM+FM), SEAL achieves even higher results than GENRE, despite the fact that this benchmark only requires page-level retrieval capabilities (that is the focus of GENRE). Finally, our intersective formulation achieves the highest results, both in hits@1 and @10, indicating that multiple ngrams identifiers might capture complementary information, which can be aggregated for stronger performances.

**Natural Questions** We report in Table 3 the results of our evaluation on Natural Questions, a passage-level retrieval benchmark with a larger collection of documents (*i.e.*, ~21M w.r.t. 200k in NQ320$k$). In this setting, the gap in performance between DSI-BART and SEAL is larger, possibly because memorizing documents identifiers in the parameters of the model becomes more challenging with larger corpora. Remarkably, the intersective formulation of SEAL achieves results comparable or superior to more established retrieval paradigms (*e.g.*, BM25, DPR and GAR), at high-recall (accuracy@100). To better understand the generalization capabilities of our retrieval solution we use the question/answer overlap split of Lewis et al. [2021b]. This study reveals that SEAL achieves the highest performance for question/answer pairs never seen during training (*i.e.*, no overlap), suggesting a better ability to generalize to completely novel questions with novel answers (*e.g.*, 3.5 points better than GAR on average).

Table 4: Retrieval results on individual KILT dev set(s), with the average in the rightmost column. Reporting passage-level R-precision (higher is better). We mark model that are also trained on additional synthetic data [Lewis et al., 2021c] with †. All SEAL models are multitask. Best among models trained only on KILT queries in bold.

| Model | FEV | T-REx | zsRE | NQ | HoPo | TQA | WoW | AVG |
|---|---|---|---|---|---|---|---|---|
| BM25 | 40.1 | 51.6 | 53.0 | 14.2 | 38.4 | 16.2 | 18.4 | 33.1 |
| DPR Maillard et al. [2021] | 43.9 | 58.5 | **78.8** | 28.1 | 43.5 | 23.8 | 20.7 | 42.5 |
| MT-DPR [Maillard et al., 2021] | 52.1 | 53.5 | 41.7 | 28.8 | 38.4 | 34.2 | 24.1 | 39.0 |
| MT-DPR [Oğuz et al., 2021] | 52.1 | **61.4** | 54.1 | 40.1 | 41.0 | 34.2 | 24.6 | 43.9 |
| MT-DPR† [Oğuz et al., 2021] | 61.4 | 68.4 | 73.3 | 44.1 | 44.6 | 38.9 | 26.5 | 51.0 |
| MT-DPR† (large) [Oğuz et al., 2021] | 62.8 | 66.6 | 66.9 | 42.6 | 42.1 | 37.9 | 23.4 | 48.9 |
| SEAL (LM+FM) | 31.5 | 42.0 | 34.0 | 21.7 | 24.7 | 21.4 | 17.6 | 27.6 |
| SEAL (LM+FM, intersective) | **67.8** | 58.9 | **78.8** | **43.6** | **54.3** | **41.8** | **36.0** | **54.5** |

Table 5: Downstream results on the KILT test set(s). Downstream metrics are accuracy (FEVER, T-REx, zero-shot RE), exact match (Natural Questions, HotpotQA, TriviaQA), or F1 (Wizard of Wikipedia). Best in bold. †: result taken from the `eval.ai` KILT leaderboard.

| System | FEV ACC | T-REx ACC | zsRE ACC | NQ EM | HoPo EM | TQA EM | WoW F1 |
|---|---|---|---|---|---|---|---|
| KGI [Glass et al., 2021][†] | 85.6 | **84.4** | 72.6 | 45.2 | - | 61.0 | 18.6 |
| Hindsight [Paranjape et al., 2021] | - | - | - | - | - | - | **19.2** |
| DPR+BART [Petroni et al., 2021] | 86.7 | 59.2 | 30.4 | 41.3 | 25.2 | 58.6 | 15.2 |
| RAG [Petroni et al., 2021] | 86.3 | 59.2 | 44.7 | 44.4 | 27.0 | 71.3 | 13.1 |
| MT-DPR+BART [Maillard et al., 2021] | 86.3 | - | 58.0 | 39.8 | 31.8 | 59.6 | 15.3 |
| MT-DPR+FiD [Piktus et al., 2021] | 89.0 | 82.5 | 71.7 | 49.9 | 36.9 | 71.0 | 15.7 |
| MT-DPR-WEB+FiD [Piktus et al., 2021] | 89.0 | 81.7 | 74.2 | 51.6 | 38.3 | **72.7** | 15.5 |
| SEAL+FiD (LM+FM) | 87.9 | 83.7 | 74.2 | 47.3 | 37.6 | 65.8 | 17.5 |
| SEAL+FiD (LM+FM, intersective) | **89.5** | 83.6 | **74.7** | **53.7** | **40.5** | 70.9 | 18.3 |

Table 6: Retrieval results on the NQ test set with different model sizes. DPR (large) performance from [Oğuz et al., 2021].

| System | # Par. | A@20 | A@100 |
|---|---|---|---|
| DPR (base) | ~220M | 80.1 | 86.1 |
| DPR (large) | ~350M | 80.2 | 86.7 |
| SEAL (large) | ~400M | 76.2 | 86.3 |

**KILT** We report retrieval results at passage level on the KILT benchmark in Table 4.[3] SEAL outperforms DPR by more than 10 points on average in passage-level R-precision, indicating that our method is more precise in surfacing ground truth evidence as the first result. Moreover, SEAL also performs better than MT-DPR (multi-task DPR) even when the latter is pretrained on tens of millions of questions from PAQ [Lewis et al., 2021c], a technique that can drastically improve results and that could potentially bring benefits to our method as well (a task we leave for future work). When it comes to downstream performances (Table 5), FiD with passages retrieved by intersective SEAL establishes a new state-of-the-art on 4 datasets out of 7 (FEVER, zsRE, NQ, HoPo), and achieves very competitive results on the remaining 3.

**Speed and constrained decoding** The inference speed of SEAL is directly proportional to the beam size, with a limited overhead added by constrained decoding. On the Natural Questions test set, for instance, retrieval with the intersective scoring requires on our 1 GPU evaluation setup ~16 minutes and ~35 minutes with, respectively, a beam size of 5 or 15. Mao et al. [2021] report a lower runtime for GAR (~5 minutes), and a comparable one for DPR (~30 minutes). Note that more efficient approaches to constrained decoding have been proposed (*e.g.*, De Cao et al. [2021a])

---

[3]We report page-level and KILT-score results in the Appendix (§C).

Table 7: Ablation on Natural Questions. SEAL when using (✓) or not using (✗) FM-index constrained decoding, for beam size values in $\{3, 5, 10, 15\}$. Reporting accuracy@$k$.

| System | Constr. | Beam | A@20 | A@100 |
|--------|---------|------|------|-------|
| SEAL | ✓ | 15 | 65.8 | 81.1 |
| (LM+FM) | ✗ | 15 | 65.3 | 80.1 |
| | ✓ | 3 | 63.3 | 78.0 |
| | ✓ | 5 | 64.7 | 79.9 |
| | ✓ | 10 | 65.4 | 80.8 |
| SEAL | ✓ | 15 | 76.2 | 86.3 |
| (LM+FM, | ✗ | 15 | 76.2 | 86.2 |
| intersective) | ✓ | 3 | 75.2 | 84.9 |
| | ✓ | 5 | 75.9 | 85.8 |
| | ✓ | 10 | 76.4 | 86.4 |

Table 8: Performance on Natural Questions with different max ngrams sizes, using SEAL (LM+FM, intersective).

| Length | A@20 | A@100 |
|--------|------|-------|
| 3 | 64.7 | 74.8 |
| 5 | 73.6 | 83.7 |
| 10 | 76.2 | 86.3 |

and we leave their application to SEAL as future work. Moreover, generation is becoming the de facto standard approach to NLP, not just as the method for materializing final outputs, but also for modeling the computational process needed before computing the answer [Wei et al., 2022]. As such, we expect generation latency will improve significantly and increasingly over time as a result of this growing interest. Any improvement will be directly applicable to SEAL.

**Model size** In Table 4 we show that parameter size is not a crucial factor behind the good performance of the method that we propose: SEAL (~400M) outperforms all model from Oğuz et al. [2021], including the MT-DPR large model (~350M) trained on NQ *and* PAQ [Lewis et al., 2021c]. In fact, the results of MT-DPR are *lower* with a bigger backbone—which could point to an increased difficulty in training larger models for dense retrieval. In the same direction, on vanilla NQ (Table 6), where DPR large only slightly outperforms DPR base, the performance of SEAL is in the same ballpark.

**Ablation studies** In Table 7 we report results on NQ for various configurations of SEAL. While, in general, performances increase with a larger beam, diminishing returns (or even a performance decrease) are found between a value of 10 and 15. Disabling constrained decoding and discarding a posteriori all generated ngrams that don't appear in the corpus results in slightly lower performances. We have found the decoding maximum length to have a crucial impact on the retrieval performance of SEAL, since shorter ngrams tend to be less informative than longer ones. We report in Table 8 performances on the NQ test set when using, respectively, 3, 5, and 10 as ngram maximum length. Every other SEAL figure reported in this paper uses 10.

**Qualitative Analysis** In Table 9, we show examples of ngrams predicted by SEAL (trained on KILT) given the query "can you predict earthquakes". SEAL is able to rephrase the query in ways that preserve its lexical material producing ngrams such as *earthquakes can be predicted*, *used to predict earthquakes* etc. Morevoer, the model is also able to explore more diverse regions of the output space, overcoming the vocabulary mismatch problem: ngrams contain related tokens like the subword *seism-* and the word *forecast*. SEAL's LM+FM scoring is also able to assign a score below 0 (and, thus, exclude from the search), unrelated ngrams that are considered by the beam because of their promising start, such as "Seismic risk in Malta @@".

Table 9: Best (top) and worst (bottom) generated keys for the query "can you predict earthquakes" (left), and retrieved documents (right). Matched ngrams in bold. "@@" separates title and body.

| score | # | identifier | doc #1 | doc #2 |
|---|---|---|---|---|
| 273.2 | 1 | earthquakes can be predicted | **Seismology** @@ for precise **earth-** **quake predictions**, including the VAN method. Most **seism**ologists do not believe that a system to pro- vide timely warnings for individual **earthquakes** has yet been developed, and many believe that such a sys- tem would be unlikely to give **useful** warning of impending **seismic** events. However, more general **forecasts** rou- tinely **predict** seismic **hazard**. Such **forecasts estimate** the **probability** of an **earthquake** of a particular [...] | **Earthquake prediction** @@ reliably identified across significant spatial and temporal scales. While part of the scientific community hold that, taking into account non-**seismic** precursors and given enough resources to study them extensively, **prediction** might be **possible**, most scientists are pes- simistic and some maintain that **earth-** **quake prediction** is inherently impos- sible. **Predictions** are deemed signif- icant if they can be shown to be suc- cessful beyond random chance.[...] |
| 272.7 | 75 | Earthquake prediction @@ | | |
| 269.9 | 3 | predicted earthquakes | | |
| 229.7 | 11 | Earthquake forecasting @@ | | |
| 217.2 | 2 | prediction Earthquake | | |
| 211.5 | 1 | used to predict earthquakes | | |
| 205.3 | 7 | earthquakes. Earthquake | | |
| − | | | | |
| −77.0 | 9 | Seismic metamaterial @@ | | |
| −97.4 | 14 | Seismic risk in Malta @@ | | |
| −113.4 | 3 | Quaternary (EP) @@ | | |
| −150.3 | 1 | used to predict the locatio[...] | | |
| −301.5 | 17 | Precipice (Battlestar Gala[...] | | |

# 7 Discussion

With SEAL we present solution that could potentially find applications outside information retrieval (*e.g.*, enforce generated substrings come from a white list of trusted sources). While we conduct our experiments with a model of ~400M parameters (*i.e.*, BART) for fast iterations, we believe the use of larger models could considerably improve performance. Changing the model would not affect the size of the index nor the cost of using it — $O(|n|\log|V|)$ for finding an ngram $n$. Moreover, we believe that indexing very large corpora (*e.g.*, the web) could be done more efficiently than existing attempts (*e.g.*, Piktus et al. [2021]) given the light memory footprint. Finally, dynamic variants [Gerlach, 2007, Salson et al., 2009] could allow the update of the FM-index on the fly without the need of re-indexing. While out of the scope of the current paper, we plan to tackle some of these scaling challenges in future work.

# 8 Conclusion

In this paper we present SEAL, a novel retrieval system that combines an autoregressive language model with a compressed full-text substring index. Such combination allows to constrain the genera- tion to existing ngrams in a corpus and to jointly retrieve documents containing them. Empirically, we show an improvement by more than 10 points in average passage-level R-precision on KILT, and establish new state-of-the-art downstream performance on 4 out 7 datasets when paired with a reader model. While our results show that SEAL could already compete with more established retrieval systems, we believe there is potential in exploring existing (or yet to come) larger autoregressive models.

# 9 Broader Impact

While our decoding methodology provides a way to enforce corpus-based generation constraints, this does not fully prevent the generation and retrieval of non-factual or abusive text, since the corpus itself may well contain non-factual or abusive text. Moreover, even if a given corpus is misinformative or abusive, the language model could still produce undesirable text by "misquoting" material from it, *e.g.*, selectively copying an utterance that is being criticized in the original context of appearance. Applications should allow user to check the original supporting text. Furthermore, while we have only used a relatively small language model, scaling up the autoregressive model of SEAL would come with the same environmental risks that any other large system would pose, due to the more demanding energy requirements.

## Acknowledgments and Disclosure of Funding

We thank Aleksandra Piktus, Edoardo Barba, Niccolò Campolungo, and Pere-Lluis Huguet Cabot for their helpful comments and suggestions.

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
