Table 10: Retrieval results on the KILT test set(s). Reporting page-level R-precision (higher is better). Best in bold. Results are taken from the `eval.ai` KILT leaderboard.

| Model | FEV | T-REx | zsRE | NQ | HoPo | TQA | WoW | AVG |
|---|---|---|---|---|---|---|---|---|
| KGI [Glass et al., 2021] | 75.6 | 74.4 | **98.5** | 63.7 | - | 60.5 | 55.4 | - |
| Hindsight [Paranjape et al., 2021] | - | - | - | - | - | - | 56.1 | - |
| GENRE [De Cao et al., 2021b] | **83.6** | **79.4** | 95.8 | 60.3 | 51.3 | **69.2** | **62.9** | **71.8** |
| MT-DPR [Maillard et al., 2021] | 74.5 | 69.5 | 80.9 | 59.4 | 42.9 | 61.5 | 41.1 | 61.4 |
| MT-DPR+WEB [Piktus et al., 2021] | 74.8 | 75.6 | 89.7 | 59.8 | 45.4 | 58.9 | 41.5 | 63.7 |
| SEAL (LM+FM) | 77.8 | 67.8 | 98.0 | 60.3 | 54.0 | 68.1 | 55.4 | 68.8 |
| SEAL (LM+FM, intersective) | 81.4 | 62.1 | 91.6 | **63.2** | **58.8** | 68.4 | 57.5 | 69.0 |

Table 11: KILT scores on the KILT test set(s). In KILT-scores an instance is considered correct if both the predicted page and the answer match the ground truth. Metrics are accuracy (FEVER, T-REx, zero-shot RE), exact match (Natural Questions, HotpotQA, TriviaQA), or F1 (Wizard of Wikipedia). Best in bold. Results are taken from the `eval.ai` KILT leaderboard.

| System | FEV K.-ACC | T-REx K.-ACC | zsRE K.-ACC | NQ K.-EM | HoPo K.-EM | TQA K.-EM | WoW K.-F1 |
|---|---|---|---|---|---|---|---|
| KGI [Glass et al., 2021] | 64.4 | **69.1** | 72.3 | 36.4 | - | 42.9 | 10.4 |
| Hindsight [Paranjape et al., 2021] | - | - | - | - | - | - | **13.4** |
| RAG [Petroni et al., 2021] | 53.5 | 23.1 | 36.8 | 32.7 | 3.2 | 38.1 | 8.8 |
| MT-DPR+BART [Maillard et al., 2021] | 63.9 | - | 50.6 | 29.1 | 9.5 | 42.4 | 5.9 |
| MT-DPR-WEB+FiD [Piktus et al., 2021] | 65.7 | 64.6 | 67.2 | 35.3 | 11.7 | 45.6 | 7.6 |
| SEAL+FiD (LM+FM) | 67.0 | 60.1 | **73.2** | 32.8 | 15.1 | 47.7 | 11.0 |
| SEAL+FiD (LM+FM, intersective) | **71.3** | 54.6 | 69.2 | **38.8** | **18.1** | **50.6** | 11.6 |

# Appendix

# A  Hyperparameters

**Training**  We finetune the model using `fairseq`. We use Adam [Kingma and Ba, 2015] with a learning rate of $3 \cdot 10^{-5}$, warming up for $500$ updates, then using polynomial decay for at $800k$ updates, evaluating every $15k$ steps. We stop the training run if the loss on the development set stops improving for $5$ evaluation passes. We use label smoothing $(0.1)$, weight decay $(0.01)$, and gradient norm clipping $(0.1)$. We train in batches of $4096$ tokens on $8$ GPUs.

**Inference**  We decode for $10$ timesteps with a beam size of $15$, and set the hyperparameters $\alpha$, and $\beta$ to, respectively, $2.0$ and $0.8$. The hyperparameters have been tuned on the Natural Questions development set. In the constrained decoding stage, we force part of the generated ngrams to match document titles.

# B  DSI-BART replication details

On NQ320$k$, `bert-base-cased` is used to compute the embeddings for the clustering. On regular NQ, we use the public precomputed DPR embeddings. To compare fairly against SEAL, we fine-tune the same encoder-decoder backbone, *i.e.*, BART large.

# C  Additional KILT results

We report in Table 10 page-level results on the KILT test set. On most datasets, SEAL obtains results which are comparable or better than other systems performing page-level retrieval. Furthermore, results are within two points of the average performance of GENRE, *i.e.*, a system that directly targets the page-level setting. Comparing KILT-scores (Table 11), *i.e.*, a metric combining downstream performances and page-level R-precision, we achieve state-of-the-art results on 4 out of 7 datasets.

Table 12: Ablation on Natural Questions. SEAL when using (✓) or not using (✗) supervised/unsupervised data. Reporting accuracy@$k$.

| System | Sup. | Unsup. | A@20 | A@100 |
|---|---|---|---|---|
| BM25 | - | - | 62.9 | 78.1 |
| SEAL | ✓ | ✓ | 76.2 | 86.3 |
| (LM+FM | ✓ | ✗ | 74.8 | 85.4 |
| intersective) | ✗ | ✓ | 61.7 | 76.3 |

## D  Impact of unsupervised examples

SEAL is trained with both supervised and unsupervised examples. In Table 12 we report ablated results, by which we assess the importance of both kind of training examples. The addition of unsupervised examples improves purely supervised training by one point (A@100). Only training with unsupervised examples results in performances which are slightly below BM25's.