# OpenReview forum: "Autoregressive Search Engines: Generating Substrings as Document Identifiers"
_NeurIPS.cc/2022/Conference — NeurIPS 2022 Accept_

### Official Review · Reviewer_wK2G · 2022-07-10

**Rating:** 6
**Confidence:** 4
**Soundness:** 3 good
**Presentation:** 3 good
**Contribution:** 3 good

**Summary:**

This paper proposed to use autoregressive language models as search engines. Previous work in this direction has explored ways to do such retrieval by generating some unique identifiers of the documents. The authors proposed an even simpler approach -- using all ngrams in a document as possible identifiers and directly generating them using an autoregressive language model. This process was done through constrained decoding with the help of FM-index. They conducted experiments on the Natural Questions and the KILT benchmark, and the experimental results demonstrated that they could achieve comparable performance with the SOTA models while using a considerably lighter memory footprint.


**Questions:**

- By comparing Table 2 in this paper with Table 3 (Tay et al., 2022), I saw that the performance varies a lot on NQ320K even for BM25. Also, they basically achieved better performance on hits@1 while you achieved better performance on hits@10. Can you elaborate on this issue a bit more? This makes me hard to judge where this work sits in the literature.
- In Table 5, I noticed that the performance of **SEAL (LM+FM, intersective)** outperforms **SEAL (LM+FM)** by a large margin (almost doubled), whereas it is not the case on NQ. Do you know why this happens?




**Limitations:**

The authors sufficiently addressed the limitations in the paper.

**Strengths And Weaknesses:**

- Originality: Although the idea of generating unique document identifiers using autoregressive models is not quite new, I do like the idea of generating the ngrams in a passage as possible identifiers. Also, I appreciate the efforts the authors put to make this simple idea work, as can be seen from the results, one needs to combine equations (2), (3), and (4) to make it really work.

- Quality: The proposed approach is technically sound and interesting. As seen from the experimental results, they outperformed the prior autoregressive baselines on NQ and KILT. They also achieved comparable performance with DPR on NQ and outperformed DPR by a large margin on KILT. Note that the authors also claimed they achieved SOTA performance on 4 out of 7 datasets on KILT. Although this does not hold by now, I don't believe it diminishes the quality of this work.
Other comments I have: (1) From my perspective, the ngram length $k$ and the number of k-grams selected during training are very important hyperparameters (set to 10 in the experiments) but lacked detailed discussion over in the paper. I would like to see an experiment regarding the performance change over a different length of ngrams. It could just be a small-scale experiment but I think having those numbers would help consolidate the claims made in the paper. (2) Correct me if I am wrong but I do not find the details of how to select a fixed number of k-grams from a total of $|d| - k$ k-grams. How important is the selecting strategy? It would be better if the authors can establish a baseline using a random selection of the k-grams.

- Clarity: In general this paper is well-organized and easy to follow. However, I think having a more formal definition of some notations in the paper could further improve the clarity. For example:
Line 142, does ngrams(K) denote $K$ ngrams or a set of k-grams?
Line 147, how is $P(n|q)$ computed?

- Significance: I think this paper studied a very interesting problem and proposed a promising approach. Although the experiments are not comprehensive for the readers to understand every aspect of the system, I think it still brings many benefits to the research community. I believe it would lead to some follow-ups in this direction.

---

> ### Author Response · Authors · 2022-08-01
> **Comment to Reviewer wK2G**
>
> Thank you for considering our approach technically sound / interesting and our paper well-organized / easy to follow. We really appreciate you believing this work can lead to follow-ups in this direction.
>
> **Ngram length**. We report here the performance on NQ with different ngram lengths. Performances go up as we increase the ngram length. We will add this table to the revised paper with a relevant discussion of the figures.
> | Model | Length | A@20 | A@100 |
> | --- | --- | --- | --- |
> | SEAL (LM+FM, intersective) | 3 | 64.7 | 74.8 |
> | SEAL (LM+FM, intersective) | 5 | 73.6 | 83.7 |
> | SEAL (LM+FM, intersective) | 10 | 76.2 | 86.3 |
>
> **How the training ngrams are chosen**. The number of training ngrams has been tuned in preliminary experiments. 10 was the optimal value on the NQ dev set, but we have to say that the effect of tweaking this parameter is not critically strong: even with the temperature parameter, biasing towards widows with high overlap with the query makes for a very peaky distribution (see our response to R#3) with usually little diversity among samples. Controlling the ngram distribution (instead of uniform random sampling) is instead very important for performances, because it pushes the model to focus towards relevant and/or predictable parts of the documents. Our method to select training ngrams is very simple, and leaves a lot of room for potential improvements. We will add an ablation study on this in the revised paper.
>
> **Mismatch reported numbers.** The authors of DSI [1] have not been able to share with us checkpoints, preprocessed datasets and document collection. We have then tried to reproduce their setup according to their specifications, but we had no way to check how well our setting matches with theirs. Therefore, the direct comparison with their reported numbers is impossible. Their BM25 numbers are computed with the `gensim` library, which seem to produce significantly lower results compared to the more commonly used Lucene bindings in the `pyserini` library, so we chose to report results obtained with both `gensim` and `pyserini`.
>
> **Notation**. Thank you for pointing out our notation was not clear enough. To answer your questions, in line 142 $K$ is just the set of ngrams generated for some query, regardless of its size; $P(n|q)$ is the probability assigned to the ngram (given the query) by the encoder-decoder itself. We will improve the clarity of that section following your suggestions.
>
> **Intersective results**. The KILT results in Table 4 use R-precision, which is almost identical to the accuracy@k with k=1. Lower values of k show more sensitivity to noise because the model assigns the same score to multiple matches: this happens not only in the KILT Table the reviewer has mentioned, but also in NQ (see Table 3). Intersective scoring partially ameliorates the issue by merging evidence for multiple matches.
>
> **References**
> * [1] Tay et al., Transformer Memory as a Differentiable Search Index, arXiv:2202.06991, 2022.

---

### Official Review · Reviewer_tLF8 · 2022-07-11

**Rating:** 8
**Confidence:** 4
**Soundness:** 4 excellent
**Presentation:** 3 good
**Contribution:** 4 excellent

**Summary:**

This work proposes SEAL, which  trains a language model to perform retrieval tasks leveraging constrained decoding over a FM-index of ngrams in a corpus. More specifically BART is finetuned to generate sampled 10 ngrams from each ground truth document biased in favor of ngrams with high character overlap with the query.

Several scoring functions are explored including:
(LM scoring) the score P(n|q) of the most probable  fixed-length ngram.
(LM+FM scoring) the pointwise mutual information between query and ngram computed from P(n|q) and P(n)
(LM+FM intersective scoring) aggregates the contribution of multiple ngrams

For page level task NQ320k, SEAL improve over the quality of a much larger retrieval solution (GENRE).  For passage level task (NQ), SEAL significantly improve over DSI, which suffers capacity losses when memorizing document ids. SEAL produces comparable results to DPR and GAR (especially for A@100 which is the most important for QA purposes). More detailed analysis shows that SEAL is a lot better on novel questions or answers.
For KILT passage retrieval task, SEAL improved the state of the art result significantly.
For KILT downstream tasks, SEAL improves state-of-the-art for 4 out of 7 tasks.

This work represents a significant progress for learned indexing structures -- combining LM with multi-point indexing and scoring.


**Questions:**

how is the ngram sampling done to  bias towards overlap with queries? there seem to be an lack of details.

why only train from the ground-truth doc? why not leverage unsupervized training for retrieval?

it seems that the model is never trained to optimize ranking quality directly. Is there any possibility for learning to rank?



**Limitations:**

see the summary

**Strengths And Weaknesses:**

see the summary

---

> ### Author Response · Authors · 2022-08-01
> **Comment to Reviewer tLF8**
>
> Thank you for considering our work “a significant progress for learned indexing structures”.
>
> **Biasing towards ngrams with high overlap**. We thank the reviewer for pointing this lack of information out. We sample training ngrams from the following distribution:
>
> $$\frac{
> e^{L(q,d_{i:i+k}) / \tau }
> }
> {
> \sum_{j=1}^{|d|-k+1} e^{L(q,d_{j:j+k})/ \tau }
> }$$
> where $L$ is the normalized Levenshtein distance between the query and a document span and $\tau = 1.5$ is a temperature parameter. We will add this information to the paper.
>
> **Unsupervised**. Similarly to [1], we have added unsupervised examples to expose the model to the full document collection (see the ablation in Appendix B). Beyond that, we have not experimented with ad hoc, large scale pretraining, as the main goal was developing a working, viable model for autoregressive retrieval. We consider unsupervised autoregressive approaches an interesting area for further research.
>
> **Possible to optimize a ranking objective?** Thanks for the suggestion! There is no reason why we could not, in principle, use a training objective that is more aligned with the scoring function. For example, we could produce a document score by summing scores of sampled ngrams, and then train the model with a contrastive objective similar to those used by dense methods. We would like to explore this as future work.
>
> **References**
> * [1] Tay et al., Transformer Memory as a Differentiable Search Index, arXiv:2202.06991, 2022.

---

### Official Review · Reviewer_L2Y4 · 2022-07-11

**Rating:** 6
**Confidence:** 5
**Soundness:** 2 fair
**Presentation:** 3 good
**Contribution:** 2 fair

**Summary:**

- The paper presents an approach (called as SEAL) for document retrieval where a language model conditioned on a question generates n-gram tokens to identify the relevant documents from the evidence documents (or passages). To enable this functionality, the method trains BART using question and n-gram pairs, where the n-grams are sampled from the gold passages. To constrain generation to output valid n-grams such that they correspond to some documents, the approach indexes the evidence documents with an efficient datastructure called FM index. The paper presents several ways to score the generated n-grams, one where the n-gram corpus level frequency is considered and another one where  they try to score multiple n-grams using intersection of tokens. They experiment with Natural Questions (NQ) Open dataset and the KILT tasks to showcase the competitiveness of their proposed approach.

**Questions:**

Please see the points under the Weaknesses section.

** EDIT **
Increasing the rating to 6.

**Limitations:**

Yes, the authors adequately addressed the limitations and potential negative societal impact of their work.

**Strengths And Weaknesses:**

**Strengths**
 - The biggest strength of the paper is the novelty of the proposed approach. While the dominant classes of retrieval methods consist of training dual encoder models with pre-computed evidence indexes, this paper presents another method with which retrieval can be performed.

- The paper is well-written, organized, sound experiments, and related work section is good. The authors have submitted the code-base which will make the work reproducible.


**Weaknesses and Suggestions for Improvements**

- Apart from NQ, the paper does not perform evaluation on popular QA datasets such as MSMARCO, TriviaQA, SQuAD, WebQuestions, and Entity Questions where the performance of baseline models are well established. These are the datasets where a large fraction of the retrieval papers report results and benchmark their models. SEAL should also train a common model on all these datasets in the DPR-Multi style to assess the generalization of their approach.

- Performance comparison with DPR and other strong dual encoder models. From the results in Table 3, SEAL falls behind DPR on Accuracy@ 5 and Accuracy@20, where its performance is considerably low. I suggest it’s important to highlight this limitation and probe further the reasons. There should be a more rigorous performance comparison with improved dual-encoder training approaches which obtain much better results. For example, the model ANCI (https://openreview.net/forum?id=zeFrfgyZln) and ICT-DPR and MSS-DPR in (https://arxiv.org/abs/2101.00408) obtains better performance numbers than DPR and these results should be reported in Table 3.

- To understand the dependence of SEAL model on number of training examples, it will be useful to compare the sample efficiency of SEAL algorithm with that of DPR on the NQ training examples.

- Considering the number of trainable parameters, where DPR trains 220M parameters while SEAL trains ~400 M parameters, the direct comparison with DPR is not fair. This should be clearly highlighted in the paper. Are the benefits of SEAL model on KILT tasks mainly due to the trainable parameter size? It would be good to finetune smaller and larger generator models such as different configurations of T5 (or T5 lm adapted models) and then study the correlation of performance vs model size.

- The writing in the method section seems a bit dull, especially the paragraphs of “factoring in FM-index frequencies” and “intersective scoring for multiple n-grams”. Using visual illustrations and diagrams to convey the information would help the reader understand the importance of these ideas more.

- Probably a minor point, I  feel that the training / inference time of SEAL / DPR should be calculated using the same compute hardware and then compared. Also, the 64 GB of DPR index size in Table 2 is when using 32 bit representation for floats. It has been shown that 16 bit or more optimized representation of passage embeddings also works just as well (https://arxiv.org/abs/2106.05346). Similarly, GPU is not always required to perform fast search. Toolkits such as ScaNN (https://arxiv.org/abs/1908.10396) also works well on CPU.

---

> ### Author Response · Authors · 2022-08-01
> **Comment to Reviewer L2Y4**
>
> Thank you for considering our paper well-written, organized, with sound experiments and a good related work section. We really appreciated you acknowledging the novelty of the proposed approach and our effort to release all code and models to reproduce our results.
>
> **Experimental setting**. The focus of our experiments is showing that autoregressive retrieval is a promising paradigm for robust retrieval in a variety of settings, rather than chasing the latest state of the art. In addition to Natural Questions, which is standard for retrieval systems to test on, we have chosen to use KILT (that includes Natural Questions and TriviaQA) as it is a well-established, diverse benchmark that features tasks that go beyond open-domain question answering: for example, it includes things like relation extraction and fact verification.
>
> **Stronger baselines**. We appreciate the reviewer’s concern with our included comparison systems. We will include stronger systems in our NQ table so as to give a better sense of what the current state of the art is, and show what the longer term goal for autoregressive retrieval should be. However, we still feel that our relevant experimental comparison system is DPR. Our goal was to establish SEAL as a simple, autoregressive baseline to be compared on equal footing against simple, dense baselines. The systems that the reviewer has mentioned offer solid improvements over the simple baseline, but do not fundamentally change the standard recipe of dense training, and it could definitely be possible to adapt and use them for autoregressive retrieval as well: ANCE improves over the dense baseline by refining negative mining; ICT-DPR and MSS-DPR boost performances by using a pretraining objective that is more aligned to retrieval.
>
> **Low result @5 and @20**. Results at low recall probably could be explained by the fact that the matching scheme does not take into account the full context, resulting in a slightly noisier ranking. In other words, non-relevant documents that happen to contain a generated ngram might be assigned the same score, with nothing to break the tie. For example, the string “the largest cat” matches both “The liger is often believed to be *the largest cat* in the world.” and “*the largest cat*amarans and monohulls also carry cars…”. Note that this is not specific to SEAL: approaches based on lexical exact matches such as BM25 or GAR are affected as well. At higher recall, e.g., @100, which is more tolerant to this kind of noise, results reach or surpass the dense retrieval baseline.
>
> **Model size**. Thanks to the reviewer for pointing this out! We already provide evidence that the number of parameters is not the main reason for the good results on KILT  in Table 4, where we outperform both DPR base and large. For completeness, we also report here a comparison with DPR large on NQ: the results @100 are slightly better, but in the same ballpark as SEAL.
> | Model | Params | A@5 | A@20 | A@100 |
> | --- | --- | --- | --- | --- |
> | DPR (*bert-base*)  | ~220M | 68.3 | 80.1 | 86.1 |
> | DPR (*bert-large*) [1]  | ~350M | 69.1 | 80.2 | 86.7 |
> | SEAL |  ~400M | 61.3 | 76.2 | 86.3 |
> We will be happy to add these numbers to the paper, as well as a discussion on the issue that the reviewer raised.
>
> **Memory usage of embeddings**. We have reported 64 GB as the size of the embedding table size for 32-bit DPR because that is what has been used to compute results that we have reported in the table. Even scaling this number down by 0.5 by using 16-bit floating points, the substance of our claim does not change, because the size of the FM-index, at 8.8 GB, is still significantly smaller.
>
> **References**
> * [1] Oguz et al., Domain-matched Pre-training Tasks for Dense Retrieval, Findings of NAACL, 2022.
> * [2] De Cao et al., Highly Parallel Autoregressive Entity Linking with Discriminative Correction, Proc. of EMNLP, 2021.

---

> > ### Comment · Reviewer_L2Y4 · 2022-08-08
> > **Follow up to the authors response**
> >
> > Thanks for providing the response!
> > Based on the response to my review and the author's responses to other reviews, I am happy to increase my score to 6.
> >
> > Some followup comments from my side that would be useful to incorporate in the next version of the paper.
> >
> > - It will still be good to provide results on the TriviaQA-Open, Squad-Open etc. datasets as was done in the DPR paper and compare them with SEAL. It will help the reader to appreciate the strengths and analyze the weaknesses of your model and will lead to progress as a whole.
> >
> > - It's good to show SOTA or near SOTA results on KILT. My only worry was that a large stream of work that came after DPR did not evaluate strongly on KILT and as a results the baselines models on KILT are not strong enough.
> >
> > -  Low result @5 and @20: It will be useful to suggest some future work directions in the discussion section in the paper that can aim to improve this.

---

> > > ### Author Response · Authors · 2022-08-10
> > > **Response to Reviewer L2Y4**
> > >
> > > Thank you for the score increase and for all the suggestions on how to strengthen the paper! We will revise the paper accordingly.

---

### Official Review · Reviewer_EwCK · 2022-07-12

**Rating:** 7
**Confidence:** 5
**Soundness:** 4 excellent
**Presentation:** 4 excellent
**Contribution:** 4 excellent

**Summary:**

This paper proposes a novel scheme to apply autoregressive language models decoding to retrieval tasks, in which documents are  represented using all constituent n-grams as possible identifiers. The key idea is to use an FM-Index to prevent the generative model from producing document identifiers with text outside of any of the indexed documents. The appeal of the approach is that large language models can be adapted (fine-tuned) for this task without major architectural changes. Empirically, the approach performs competitively with a selection of recent baselines, in some cases outperforming them.



**Questions:**

* What are the implications of the assumption that documents are indexed by exact n-grams? For example, it would be good to provide intuition for why this does (or does not) preclude matching documents via synonyms or related words / expressions.

**Limitations:**

* I would have liked to see more discussion of inference-time latency.


**Strengths And Weaknesses:**

This paper does a good job explaining current drawbacks of autoregressive models in retrieval tasks (related work) and the empirical results demonstrate that the proposed scheme does improve the quality of retrieval results.

My main concern is that there may be better ways to apply large language models for retrieval, such as the re-ranking approach that is mentioned in the introduction; this wouldn't require complex generation involving indices and constrained decoding. Indeed, re-ranking shares the purported benefits of the proposed approach, namely straightforward application of large language models without architectural changes, with the key advantage that it only requires *scoring* a (query, target) pair. Autoregressive models, by definition, suffer from slow decoding speed, and therefore I'm skeptical the latency would be low enough for this approach to be practical in many settings.

Regarding the experiments, not enough motivation is given for why the baselines were selected, and it's not clear whether these truly represent SoTA results. For example, do you compare to a re-ranking baseline? There are many approaches from the IR community such as ColBERT that seem relevant but are not discussed.

Regarding the selected baselines, it doesn't look like the baselines leverage any sort of constrained decoding. It would be interesting to constrain decoding using simpler alternatives to fully understand the impact of the FM index and proposed weighting scheme.

The paper is readable and well-written. Additionally, the authors detail their experiments well and share their code which bodes well for reproducibility.

L37: extend->extent
L230: "our the size of our index is"

---

> ### Author Response · Authors · 2022-08-01
> **Comment to Reviewer EwCK**
>
> Thank you for defining our paper readable and well-written, for considering the drawbacks of autoregressive models well explained, for defining our approach appealing and for acknowledging that the empirical results demonstrate that our solution can improve the quality of retrieval results.
>
> **Re-ranking**: the paper describes a first-stage retrieval system to get candidates from a large collection of documents. Current LM-based reranking solutions can be applied only to a small set of documents given the computational cost of cross encoding query and document text, so they are not well suited as first-stage retrieval solutions. A re-ranking step is orthogonal to our solution and can be applied to the top-k results from SEAL to boost retrieval results.
>
> **Latency**. The main bottle for SEAL is, as you have suggested, decoding speed rather than the search itself, since the FM-index’s querying time complexity does not depend on corpus size. Decoding speed, however, can be greatly improved. Increasing decoding speed is a hot area of research and there are several works that propose solutions to speedup the process (e.g., [1], https://huggingface.co/blog/tf-xla-generate). Moreover, generation is becoming the de facto standard approach to NLP, not just as the method for materializing final outputs, but also for modeling the computational process needed before computing the answer (think chain-of-thought, workspaces etc.). As such, we expect generation latency will improve significantly and increasingly over time as a result of this growing interest. Any improvement will be directly applicable to SEAL. We will add a discussion on this issue in the revised paper!
>
> **Stronger baselines**. We make SoTA claims w.r.t. the public KILT leaderboard where, at the time of writing the paper, SEAL was outperforming all competing systems on some datasets. However, our aim is not chasing SoTA but presenting a novel approach to retrieval that achieves comparable results to other families of solutions (e.g., dense), which may, with additional research and development, ultimately outperform other approaches.
>
> **Constrained decoding**. We have an ablation in Table 6: constrained decoding improves performance compared to the unconstrained baseline. However, the unconstrained baseline still produces good results, as non-occurring ngrams get filtered out anyways (as they produce no matches).
>
> **Vocabulary mismatch/synonyms**. Although documents are indexed by exact n-grams, the NLG capabilities of language models can avoid vocabulary mismatch by generating multiple n-grams containing synonyms or related concepts from multiple documents (Table 7 reports an example of this behavior).
>
> **References**
> * [1] De Cao et al., Highly Parallel Autoregressive Entity Linking with Discriminative Correction, Proc. of EMNLP, 2021.

---

### Meta-Review · Area_Chair_9YQ8 · 2022-08-25

**Recommendation:** Accept
**Confidence:** Certain

**Metareview:**

This paper proposes a method (SEAL) for document retrieval where a language model (LM) conditioned on a question generates n-grams as document identifiers. This is done by training BART on question and n-gram pairs, where the n-grams are sampled from the gold passages, and at test time constraining generation to output valid n-grams that correspond to document identifiers. Experiments on Natural Questions (NQ) Open dataset and the KILT tasks obtain strong results.

Overall, all reviewers agree that this is a strong paper that proposes a simple but effective approach. I agree with their assessments and recommend acceptance. However, a weakness that has been pointed out is that the paper does not perform evaluation on other common QA benchmarks (MSMARCO, TriviaQA, SQuAD, WebQuestions, and Entity Questions) where the performance of baseline models are well established. I strongly encourage the authors to train SEAL on at least some of those datasets and compare with stronger baselines.

**Award:**

No

---

### Decision · Program_Chairs · 2022-09-14

Accept